# Improving Prototypical Part Networks with Reward Reweighing, Reselection, and Retraining

## Abstract

In recent years, work has gone into developing deep interpretable methods for image classification that clearly attributes a model's output to specific features of the data. One such of these methods is the *prototypical part network* (ProtoPNet) (Chen et al., 2019), which attempts to classify images based on meaningful parts of the input. While this method results in interpretable classifications, it often learns to classify from spurious or inconsistent parts of the image. To address this problem, we take inspiration from the recent developments in Reinforcement Learning with Human Feedback (RLHF) to detect and debug low-quality prototypes. We propose the *reweighing, reselecting, and retraining* (R3) debugging framework, which performs three additional corrective updates to a pretrained ProtoPNet in an offline and efficient manner. The first two steps are reward-based reweighing and reselection, which involve learning a reward model based on collected human feedback and then making the prototypes align with human preferences. The final step is retraining, which realigns the base features and the classifier layer of the original model with the updated prototypes. We find that in most cases our R3 framework consistently improves both the interpretability and the predictive accuracy of ProtoPNet.

## 1 Introduction

With the widespread use of deep learning, having these models be interpretable is more important now than ever. As these models continue to see use in high-stakes situations, practitioners hoping to justify a decision need to understand how a deep model makes a prediction, and trust that those explanations are valuable and correct (Rudin et al., 2021). One such proposed method for image classification is the *prototypical part network* (ProtoPNet), which classifies a given image based on its similarities to prototypical parts of different classes, called prototypes (Chen et al., 2019). This model aims to combine the power of deep learning with an intuitive reasoning module similar to humans.

While ProtoPNet aims to learn meaningful prototypical concepts, in practice, learned prototypes suffer from learning spurious concepts, such as the background of an image, from inconsistent concepts, such as learning both the head and the wing of a bird, and from duplicating concepts, such as having two prototypes that correspond to the same wing of the same bird (Bontempelli et al., 2023). Such problems are highly detrimental to the efficacy of these models, resulting in wasted computation at best and incorrect reasoning at worst. Various methods have been proposed to account for these issues (Bontempelli et al., 2023; Nauta et al., 2021; Barnett et al., 2021), but these methods involve either costly online labelling procedures or fall short of providing a persistent means of measuring prototype quality.

We seek to increase the performance of the learned prototypes by taking inspiration from recent advances in reinforcement learning with human feedback (RLHF) (Ouyang et al., 2022) and reward learning (Lee et al., 2023). RLHF and reward learning have become popular approaches for aligning large language models with human preferences, partially due to the flexibility of learned rewards

---

All source code will be released upon acceptance

and feedback collection methods (Askell et al., 2021). While prior work has incorporated human feedback into ProtoPNets (Barnett et al., 2021; Bontempelli et al., 2023), no variation of ProtoPNet has incorporated a cheap and flexible reward-based debugging framework.

zTowards this end, we propose the *reward reweighed, reselected, and retrained prototypical part network* (R3-ProtoPNet), which seeks to improve the original ProtoPNet via debugging with a learned reward model. With limited human feedback data on the Caltech-UCSD Birds-200-2011 (CUB-200-211) dataset (Welinder et al., 2010), we are able to train a high-quality reward model that achieves $90.1\%$ test accuracy when ranking human preferences, serving as a strong measure for prototype quality. Two distinct advantages of having an external reward model that faithfully captures human preferences are: 1) the debugging process becomes efficient, because it doesn't require online feedback as the reward model is pretrained; and 2) the objective could be flexible, as different reward models could represent slightly different preferences. Empirically, R3-ProtoPNet is able to improve the meaningfulness of prototypes, removing dependence on spurious features, and is able to slightly decrease inconsistency across images compared to the original ProtoPNet. When used either individually or as base learners in an ensemble, R3-ProtoPNet is generally able to outperform the original ProtoPNet on a held-out test dataset in terms of predictive accuracy.

The contributions of this work can be summarized as follows:

- We demonstrate that a reward model trained on efficiently collected human feedback data can accurately rank human preferences.

- We propose using the robust reward model as a quantified metric of prototype quality and model interpretability.

- We introduce the R3 framework and R3-ProtoPNet, which uses efficient reward-guided debugging to improve prototype meaningfulness and predictive performance.

## 2    RELATED WORK

### 2.1    REINFORCEMENT LEARNING WITH HUMAN FEEDBACK

Since the success of InstructGPT (Ouyang et al., 2022), Reinforcement Learning with Human Feedback (RLHF) has received a great deal of attention in the machine learning community. Although this success is recent, incorporating human feedback into reinforcement learning methods via a learned reward model has a deep history in reward learning (Christiano et al., 2017; Jeon et al., 2020). While works taking inspiration from InstructGPT have used proximal policy optimization (PPO) to fine-tune networks with human feedback (Bai et al., 2022), it is unclear to the extent that formal reinforcement learning is necessary to improve models via learned reward functions (Lee et al., 2023), or if the human feedback needs to follow a particular form (Askell et al., 2021). Some prior work incorporates the reward function as a way to weigh the likelihood term (Stiennon et al., 2022; Ziegler et al., 2019). Keeping this work in mind, we incorporate the reward model into ProtoPNet as a way to reweigh prototypes post-training.

### 2.2    EXAMPLE-BASED MODELS AND PROTOTYPICAL PART NETWORKS

The field of interpretable deep learning is vast, with a plethora of explainability and interpretability methods available to the user. For a more complete overview of interpretable deep learning, please refer to Rudin et al. (2021). To ground the discussion, we focus primarily on example-based models, one such example being ProtoPNet. While ProtoPNet is our model of interest, other example-based methods exist, such as the non-parametric xDNN (Angelov and Soares, 2019) or SITE, which performs predictions directly from interpretable prototypes (Wang and Wang, 2021). While other example-based methods exist, we focus on the ProtoPNet due to its intuitive reasoning structure.

Since its introduction by Chen et al. (2019), ProtoPNets have received a great deal of attention, and various iterations have been developed. Work has explored extending the ProtoPNet to different architectures such as transformers (Xue et al., 2022), or sharing class information between prototypes (Rymarczyk et al., 2021). Donnelly et al. (2022) increase the spatial flexibility of ProtoPNet, allowing prototypes to change spatial positions depending on the pose information available in the image.

ProtoPNets and variations have seen success in high-stakes applications, such as kidney stone identification (Flores-Araiza et al., 2022) and mammography (Barnett et al., 2021).

Many works have commented on how the original ProtoPNet tends to overemphasize spurious features, and they have taken different approaches to solving this issue. Nauta et al. (2021) introduce a explainability interface to ProtoPNet, allowing users to see the dependence of the prototype on certain image attributes like hue and shape. The authors claim that seemingly dissimilar or spurious prototypes share certain difficult-to-perceive features, like texture or contrast. Barnett et al. (2021) introduce a variation of the ProtoPNet, IAIA-BL, which biases prototypes towards expert labelled annotations of classification-relevant parts of the image.

Similar to how we provide human feedback at the interpretation level, Bontempelli et al. (2023) introduce the ProtoPDebug, where a user labels a prototype and image pair as "forbidden" or "valid", and a fine-tuning step maximizes the distance between learned prototypes and patches in the forbidden set and minimizes the distance between learned prototypes and patches in the valid set. While also incorporating human feedback, Bontempelli et al. (2023) do not ground their method in RLHF, but instead includes the binary feedback as a supervised constraint into the ProtoPNet loss function. Learning a reward function via ratings allows us to simultaneously increase the interpretability of the prototypes, and develop an evaluation metric for the quality of a particular prototype. Compared to previous approaches, our R3 framework allows for fast collection of high-quality human feedback data and the construction of a reward model that measures prototype quality while increasing the interpretability and the predictive performance of the model.

## 3 PROTOTYPICAL PART NETWORK (PROTOPNET)

In this section, we describe the base architecture used in our method, the Prototypical Part Network (ProtoPNet) introduced in Chen et al. (2019). The ProtoPNet aims to introduce interpretability to otherwise uninterpretable image classifiers. In place of predicting from an arbitrary representation, the model makes a classification based on part attention and similar prototypical parts of an image. The general reasoning of a model is to classify an unseen image by finding training images with similar prototypical parts to those of the unseen image. This approach allows the user to interrogate the reasoning of the model, and clearly see which parts of the image led to the model's classification.

### 3.1 DESCRIPTION

Here we briefly describe the ProtoPNet, adopting the notation used in Chen et al. (2019). The ProtoPNet architecture builds on a base convolutional neural network $f$, which is then followed by a prototype layer denoted $g_p$, and a fully connected layer $h$. Typically, the convolutional features are taken pretrained models like VGG-19, ResNet-34, or DenseNet-121.

The ProtoPNet injects interpretability into these convolutional architectures with the prototype layer $g_p$, consisting of $m$ prototypes $\boldsymbol{P} = \{p_j\}_{j=1}^m$ typically of size $1 \times 1 \times D$, where $D$ is the shape of the convolutional output $f(x)$. By keeping the depth the same as the output of the convolutional layer, but restricting the height and width to be smaller than that of the convolutional output, the learned prototypes select a patch of the convolutional output. Reversing the convolution leads to recovering a prototypical patch of the original input image $x$. Using upsampling, the method constructs a activation pattern per prototype $p_j$.

To use the prototypes to make a classification given a convolutional output $z = f(x)$, ProtoPNet's prototype layer computes a max pooling over similarity scores: $g_{p_j}(z) = \max_{\tilde{z} \in \text{patches}(z)} \log((\|\tilde{z} - p_j\|_2^2 + 1)(\|\tilde{z} - p_j\|_2^2 + \epsilon))$, for some small $\epsilon < 1$. This function is monotonically decreasing with respect to the distance, with small values of $\|\tilde{z} - p_j\|_2^2$ resulting in a large similarity score $g_{p_j}(z)$. Assigning $m_k$ prototypes for all $K$ classes, such that $\sum_{k=1}^K m_k = m$, the prototype layer outputs a vector of similarity scores that matches parts of the latent representation $z$ to prototypical patches across all classes. The final layer in the model is a linear layer connecting similarities to class predictions.

In order to ensure that the prototypes match specific parts of training images, during training the prototype vectors are projected onto the closest patch in the training set. For the final trained ProtoPNet, every $p_j$ corresponds to some patch of a particular image.

## 3.2 LIMITATIONS OF PROTOPNET

While ProtoPNet is capable of providing interpretable classifications, the base training described in (Chen et al., 2019) results in prototypes that are inconsistent and represent spurious features of the image (Barnett et al., 2021; Bontempelli et al., 2023).

Chen et al. (2019) note that a prototype whose top $L$ (usually $L = 5$) closest training image patches come from different classes than the target class tend to be spurious and inconsistent, focusing on features like the background. To remedy this issue, they introduce a pruning operation, removing these prototypes entirely. While pruning does remove dependency on some subpar prototypes, we find that pruning still leaves some prototypes that rely on spurious and inconsistent features and does not improve accuracy due to substantial information loss. We visualize subpar prototypes in Figure 1. For more examples of low-quality prototypes, please see the Appendix.

## 4 HUMAN FEEDBACK AND THE REWARD REWEIGHED, RESELECTED, AND RETRAINED PROTOTYPICAL PART NETWORK (R3-PROTOPNET)

Inspired by the recent advances in reinforcement learning with human feedback (RLHF) (Ouyang et al., 2022), the reward reweighed, reselected, and retrained prototypical part network (R3-ProtoPNet) utilizes a learned reward model to fine-tune prototypes. In place of pruning prototypes and sacrificing potential information, we demonstrate that incorporating human feedback into the training of the ProtoPNet improves prototype quality while increasing ensemble accuracy. In this section, we describe the collection of high-quality human feedback data, our reward model, and how we incorporate the reward model into the debugging loop via a three-step update procedure.

## 4.1 HUMAN FEEDBACK COLLECTION

A crucial aspect behind the success of RLHF methods is the collection of high quality human feedback data. Unclear or homogeneous feedback may result in a poor performing reward model (Christiano et al., 2017). The design of human feedback collection is vitally important to the training of a useful reward model.

The inherent interpretability of ProtoPNet is particularly useful for RLHF. Given a trained ProtoPNet, it is possible for a user, who doesn't have to be an expert, to directly critique the learned prototypes. Given a particular classification task, a human rater should be able to recognize if a particular prototype is "good" or "bad" (Bontempelli et al., 2023). In the case of classifying birds in the CUB-200-2011 dataset, it is clear that if a prototype gives too much weight to the background of the image (spurious), or if the prototype corresponds to different parts of the bird when looking at different images (inconsistency), so that the learned prototype is not meaningfully or interpretably contributing to prediction. Given these prototypes that fail to contribute to prediction, a lay person trying to classify birds would be able to rate these prototypes as "bad", with a proper rating rubric.

There are many different ways to elicit this notion of "goodness" from a user (Askell et al., 2021). Although it is possible to incorporate many different forms of feedback into the R3-ProtoPNet, such as asking a user to compare prototypes to elicit preferences or ask for a binary value of whether a prototype is "good" or "bad", we found most success with asking the users to rate a prototype on a scale from 1 to 5. While scalar ratings can be unstable across different raters, with a clear, rule-based rating method, rating variance is reduced and it is possible to generate high-quality labels. An example rating scale on the CUB-200-2011 dataset is provided in Figure 1.

## 4.2 REWARD LEARNING

We note that, when a user provides feedback on a prototype, it is not the training image or the model prediction that the user is providing feedback on, but the prototype's resulting interpretation: the activation patterns. Our task is therefore different from RLHF applied to language modeling or RL tasks (Ouyang et al., 2022; Christiano et al., 2017), where human feedback is provided on the model output or resulting state. We therefore collect a rating dataset $\mathcal{D} = \{(x_i, y_i, h_{i,j}, r_{i,j})\}_{i=1,j=1}^{n,m}$, where $x_i, y_i$ are the training image and label, and $h_{i,j}, r_{i,j}$ are prototype $p_j$'s activation patterns

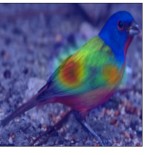 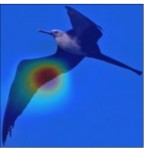 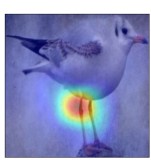 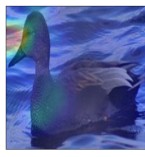 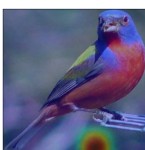

**5 - 80%-100%**
**overlap with bird**

**4 - 50%-80%**
**overlap with bird**

**3 - 10%-50%**
**overlap with bird**

**2 - 0%-10%**
**overlap with bird**

**1 - No overlap**
**with bird**

Figure 1: Rubric used for human feedback on the activation patterns of predictions for birds from the CUB-200-2011 dataset. First, the rater estimates a base score based on overlap proportion, and then an optional adjustment $\delta \in \{-1, 1\}$ could be given based on how characteristic the focused body part is.

and user-provided ratings for image $x_i$. We note that collecting preferences for this entire dataset is prohibitive and unnecessary, so we only collect a subset.

Given the dataset $\mathcal{D}$, we generate the induced comparison dataset, whereby each entry in $\mathcal{D}$ is paired with one another. Given $i \neq i'$ and/or $j \neq j'$, we populate a new paired dataset, $\mathcal{D}_{paired}$, which consists of the entries of $\mathcal{D}$ indexed by $i, j, i', j'$, and a comparison $c$, which takes values $-1, 1$. If the left-hand sample is greater, and therefore considered higher-quality, $r_{i,j} > r_{i',j'}$, then $c = -1$. If the right-hand sample is greater $r_{i,j} < r_{i',j'}$, then $c = 1$. We note that, during learning, we exclude entries with $|r_{i,j} - r_{i',j'}| < 0.5$ to increase the contrast between pairs. This synthetic construction allows us to model the reward function, $r(x_i, h_{i,j})$ via the Bradley-Terry Model (Bradley and Terry, 1952), which has demonstrated success in learning pairwise user preferences (Christiano et al., 2017). We train this model with the same loss function as in Christiano et al. (2017), a cross-entropy loss over the probabilities of ranking one pair over the other (See Appendix B for details). This synthetic construction combinatorially increases the amount of preference data, allowing us to train a high-quality reward model on relatively small amounts of human feedback data.

### 4.3 REWARD REWEIGHED, RESELECTED, AND RETRAINED PROTOTYPICAL PART NETWORK (R3-PROTOPNET)

After having collected high-quality human feedback data and trained a reward model, we can now incorporate it into a fine-tuning framework to improve the interpretability of ProtoPNet. We incorporate the reward model via a three step process consisting of reward weighting, reselection, and retraining. Each step is described in more detail below. We also include an illustrative figure explaining the framework in the Appendix.

#### 4.3.1 REWARD REWEIGHING

Although PPO is a popular option for RLHF (Ouyang et al., 2022), there is evidence that simpler fine-tuning algorithms can lead to similar performance increases (Askell et al., 2021). Inspired by the success and the efficiency of reward-weighted learning (Lee et al., 2023; Stiennon et al., 2022; Ziegler et al., 2019), we develop a reward-weighted update for the ProtoPNet:

$$\max_{p_j} \mathcal{L}_{reweigh}(z_i^*, p_j) = \max_{p_j} \sum_{i \in I(p_j)}^{n} r(x_i, p_j) \frac{1}{\lambda_{dist} \|z_i^* - p_j\|_2^2 + 1} \tag{1}$$

where $z_i^* = \text{argmin}_{z \in \text{patches}(f(x_i))} \|z - p_j\|_2^2$, $I(p_j) = \{i \mid y_i \in \text{class}(p_j)\}$, and $\lambda_{dist}$ is a fixed hyperparameter. We note that the objective function $\mathcal{L}_{reweigh}$ is a sum of the inverse distances weighted by the reward of the prototype on that image. Since we only update the prototype $p_j$, the only way to maximize this objective is to minimize the distance between prototype and image

patches with high reward $r(x_i, p_j)$. This causes the prototype to resemble high reward image patches, improving the overall quality of the prototypes. Wanting to preserve prototypes that already have high reward, we only update those prototypes that have relatively low mean reward $\gamma$, and choose this reweighing threshold per base architecture and reward model (see the Appendix for threshold choices). $\lambda_{dist}$ is included in the objective function to rescale distances, since the closest distances are near zero. We find best performance with $\lambda_{dist} = 100$.

Practically, we find that optimizing this objective function leads to locally maximal solutions, resulting in local updates that do not modify prototypes with low quality values of 1, but it's more likely to improve prototypes with quality values of 2 or higher. If the prototype $p_j$ has high activation over the background of an image $x_i$, for example, the closest patches $z_i^*$ in the training data will also be background patches, and the reward of the prototype will be low, leaving minimal room for change. It is not possible for this update to dramatically change the location of the patch in the image via this loss function.

### 4.3.2 PROTOTYPE RESELECTION

In order to improve low quality prototypes that require significant manipulation, we introduce a reselection procedure based on a reward threshold. Given a prototype $p_j$ and image $x_i$, if $\frac{1}{n_k}\sum_{i \in I(p_j)} r(x_i, p_j) < \alpha$, where $\alpha$ is a pre-determined threshold and $n_k$ is the number of training images in class $k$, or if the patch of the given prototype $p_j$ matches the patch of another prototype of the same class, we reselect the prototype. The reselection process involves iterating over patch candidates $z_i'$ and temporarily setting the prototype $p_j' = z_i'$, where $z_i'$ is chosen randomly from the patches of a randomly selected image $x_i'$ in the class of $p_j$. If $\frac{1}{n_k}\sum_{i \in I(p_j)} r(x_i', p_j') > \beta$, where $\beta$ is an acceptance threshold, and if none of the prototypes match patch $p_j' = z_j'$, then we accept the patch candidate as the new prototype. We found that varying the $\alpha$ and $\beta$ values per base architecture led to the best performance (See the Appendix for threshold choices). We refer to the combination of reweighting and reselection as the R2 update step, and the corresponding trained model the R2-ProtoPNet.

The reasoning process behind our prototype reselection method takes inspiration from the original push operation in Chen et al. (2019). Similar to how ProtoPNet projects prototypes onto a specific training image patch, here we reselect prototypes to be a particular reward-filtered training image patch. With a high enough acceptance threshold $\beta$, this forces the elimination of low reward prototypes while preserving the information gain of having an additional prototype.

One possible alternative approach is to instead search over the training patches, and select those patches with the highest reward. We found that randomly selecting patches, in place of searching for patches with the highest reward, led to higher prototype diversity and less computation time. As discussed in Section 7, it is possible that a reward model that more explicitly accounts for prototype diversity could alleviate the duplicate issue, but we leave this to future work.

While we do not use a traditional reinforcement learning algorithm to fine-tune our model as is typically done in RLHF (Askell et al., 2021), pairing the reselection and fine-tuning steps together resembles the typical explore-exploit trade-off in RL problems. We see that fine-tuning with our reward model leads to exploit behavior, improving upon already high-quality prototypes. At the same time, the reselection step serves as a form of exploration, drastically increasing the quality of uninformative prototypes. We find that these similarities are enough to improve the quality of ProtoPNet, as discussed in the next section.

### 4.3.3 RETRAINING

A critical step missing in the R2 update is a connection to prediction accuracy. As discussed in Section 5, without incorporating predictive information, performing the reward update alone results in lowered test accuracy. Since the above updates only act on the prototypes themselves, not the rest of the network, the result is a misalignment between the prototypes and the model's base features and final classifier layer. The reward update guides the model towards more interpretable prototypes, but the reward update alone fails to use the higher quality prototypes for better prediction.

To account for the lack of predictive performance, the final step of R3-ProtoPNet is retraining. Simply retraining with the same loss function used in the original ProtoPNet update results in the realignment

of the prototypes and the rest of the model. Although one could worry that predictive accuracy would reduce the interpretability of the model (Rudin et al., 2021), we find that retraining increases predictive accuracy while maintaining the quality increases of the R2 update. The result is a high accuracy model with higher-quality prototypes. We explore evidence of this phenomenon and why this is the case in the following section.

## 5 EXPERIMENTS

Here we discuss the results of training the R3-ProtoPNet on the CUB-200-2011 dataset, the same dataset as used in Chen et al. (2019). We demonstrate that the R3-ProtoPNet leads for higher quality prototypes across base model architectures and prototype configurations while not sacrificing predictive performance.

### 5.1 DATASETS

R3-ProtoPNet requires the original dataset for the initial training, as well as additional scalar ratings of the selected activation patterns produced by image-prototype pairs. Combined, this results in the dataset described in Section 4. To offer better comparison against the original ProtoPNet, we use the same dataset for initial training that was used in Chen et al. (2019), the CUB-200-2011 dataset (Wah et al., 2011). The CUB-200-2011 dataset consists of roughly 30 images of 200 different bird species. We employ the same data augmentation scheme used in Chen et al. (2019), which adds additional training data by applying a collection of rotation, sheer, and skew perturbations to the images, resulting in a larger augmented dataset.

For the collection of the activation pattern ratings, we only provided the activation patterns overlaid on the original images to the raters. Using Amazon Mechanical Turk to recruit six workers per prototype-image, we took the average as the user-provided rating for that pair. In total, 700 rated prototype-image pairs are collected according to the scale approach described in Figure 1, and we randomly selected 500 of them (the rest were used as a held-out test set to evaluate the robustness) to train the reward model.

### 5.2 ARCHITECTURES AND TRAINING

Similar to Chen et al. (2019), we study the performance of R3-ProtoPNet across five different base architectures: VGG-19, ResNet-34, ResNet-50, DenseNet-121, and DenseNet-161. While the original ProtoPNet sets the number of prototypes per class at $m_k = 10$, we additionally run the VGG-19 architecture with $m_k = 5$ prototypes to explore model performance when the number of prototypes is limited. No other modifications were made to the original ProtoPNet architecture. We train for 50 epochs and report results for the best performing model.

The reward model $r(x_i, h_i)$ is similar to the base architecture of the ProtoPNet. Two ResNet-50 base architectures take in the input image $x_i$ and the associated acticvation pattern $h_i$ separately, and both have two additional convolutional layers. The outputs of the convolutional layers are concatenated and fed into a final linear layer with sigmoid activation to predict the Bradley-Terry ranking. Predicted rewards are therefore bound in the range $(0, 1)$. We train the reward model for 5 epochs on a synthetic comparison dataset of 49K paired images and preference labels derived from 500 human ratings, and evaluate on 14K testing pairs. The reward model achieves 90.09% test accuracy. We additionally analyze the sensitivity of the reward model and R3-ProtoPNet to the amount of human feedback used for reward model training (see Appendix D), and the results suggest that the performance gain of R3-ProtoPNet can be achieved with even fewer human ratings (around 300 image-prototype pairs).

### 5.3 EVALUATION METRICS

To evaluate the performance of R3-ProtoPNet, we compare it to ProtoPNet on three metrics: test accuracy, reward, and activation precision (AP). We use test accuracy to measure the predictive performance of the models. As the above section demonstrates, the learned reward model achieves high accuracy in predicting which prototype ranks above another in accordance with human preferences, so we therefore use it as a measure of prototype quality. The final metric, activation precision, is a common metric that has been used in prior work to evaluate the overlap between a prototype's

activations and the pixels associated with a given bird (Barnett et al., 2021; Bontempelli et al., 2023), which provides a metric of interpretability independent of our method. In our work, we report a modified version of AP introduced in Bontempelli et al. (2023) to consider the specific value of the activation at each single pixel, not just the overlap alone.

## 5.4 Results

After training ProtoPNet, running the R2 update step, and then performing retraining, we see several trends across multiple base architectures. In Table 1, we report the test accuracy of the different base architectures across stages of R3-ProtoPNet training. Generally, the test accuracy from ProtoPNet temporarily decreases after applying the R2 update, but retraining could effectively recover the predictive loss, either maintaining or improving test accuracy.

In Table 2, we report the average reward of all prototypes on all test images for a given base architecture. We see that ProtoPNet achieves an average reward between 0.32 and 0.58 across architectures, with R3-ProtoPNet increasing average reward across all base architectures. In total, R3-ProtoPNet achieves a 31.65% increase in average reward across all base architectures over ProtoPNet.

Finally, we report activation precision in Table 3. Across all base architectures, ProtoPNet has an average activation precision of 43.18. As performing the R2 updates increases the average reward for all base architectures, we observe a similar, if not more substantial, increase in activation precision. Additionally, the final R3 step actually continues to improve activation precision, suggesting that further training increases attention on the bird even further. In all, R3-ProtoPNet results in an 46.85% increase in average activation precision over the base ProtoPNet.

| Base ($m_k$) | ProtoPNet | R2-ProtoPNet | R3-ProtoPNet |
|---|---|---|---|
| VGG-19 (5) | $73.35 \pm 0.16$ | $62.54 \pm 1.38$ | $\mathbf{75.54} \pm 0.19$ |
| VGG-19 (10) | $73.78 \pm 0.23$ | $47.61 \pm 1.65$ | $\mathbf{75.34} \pm 0.27$ |
| ResNet-34 (10) | $77.82 \pm 0.10$ | $53.44 \pm 2.74$ | $\mathbf{78.80} \pm 0.16$ |
| ResNet-50 (10) | $76.53 \pm 0.16$ | $59.14 \pm 2.36$ | $\mathbf{76.75} \pm 0.19$ |
| DenseNet-121 (10) | $75.64 \pm 0.22$ | $52.31 \pm 2.30$ | $\mathbf{77.10} \pm 0.15$ |
| DenseNet161 (10) | $\mathbf{78.21} \pm 0.25$ | $68.75 \pm 2.41$ | $78.04 \pm 0.30$ |
| Ensemble (VGG-19(10) + ResNet-34 + DenseNet-121 + DenseNet-161) | $80.95 \pm 0.11$ | $76.26 \pm 0.84$ | $\mathbf{82.86} \pm 0.16$ |

Table 1: R3 updates tend to increase the test accuracy. Average accuracies and standard deviations are reported across five runs, where $m_k$ is the number of prototypes per class.

| Base ($m_k$) | ProtoPNet | Reselected | Reweighed | R3-ProtoPNet |
|---|---|---|---|---|
| VGG19 (5) | 0.58 | 0.64 | 0.67 | 0.71 |
| VGG19 (10) | 0.41 | 0.57 | 0.64 | 0.65 |
| ResNet-34 (10) | 0.39 | 0.45 | 0.50 | 0.48 |
| ResNet-50 (10) | 0.32 | 0.43 | 0.47 | 0.45 |
| DenseNet-121 (10) | 0.48 | 0.53 | 0.54 | 0.58 |
| DenseNet-161 (10) | 0.43 | 0.51 | 0.57 | 0.54 |
| Average | 0.436 | 0.522 | 0.564 | 0.574 |

Table 2: R3-ProtoPNets significantly outperform the original ProtoPNets in terms of the image-prototype rewards estimated by our reward model. Values are averaged over the entire test dataset. We divide the R2 update into two columns **Reselected** and **Reweighed** to better show individual effect of each step. By construction, values lie on the interval $[0, 1]$.

## 5.5 Discussion

Given the results, we can conclude that incorporating reward information into the ProtoPNet via the R3 updates does increase its interpretability while maintaining or improving predictive performance,

| Base ($m_k$) | ProtoPNet | Reselected | Reweighed | R3-ProtoPNet |
|---|---|---|---|---|
| VGG19 (5) | $56.25 \pm 0.67$ | $60.39 \pm 0.98$ | $66.31 \pm 1.92$ | $70.25 \pm 1.21$ |
| VGG19 (10) | $50.08 \pm 0.64$ | $57.12 \pm 1.34$ | $70.27 \pm 2.42$ | $68.37 \pm 1.45$ |
| ResNet-34 (10) | $32.04 \pm 0.28$ | $39.88 \pm 1.42$ | $52.93 \pm 2.15$ | $59.21 \pm 2.40$ |
| ResNet-50 (10) | $38.69 \pm 0.49$ | $41.09 \pm 1.85$ | $46.26 \pm 3.66$ | $55.52 \pm 1.94$ |
| DenseNet-121 (10) | $36.49 \pm 0.37$ | $52.21 \pm 1.27$ | $62.94 \pm 3.23$ | $66.74 \pm 0.86$ |
| DenseNet-161 (10) | $45.52 \pm 0.78$ | $51.44 \pm 1.17$ | $60.87 \pm 2.24$ | $60.36 \pm 1.28$ |
| Average | 43.18 | 50.36 | 59.93 | 63.41 |

Table 3: Average Activation Precision (AP) over the test dataset are increased across different stages of R3 updates. Standard deviation across the five runs is reported.

and there's no such long-term trade-off between interpretability and accuracy. There's also strong correlation between the reward values and the AP metric, suggesting the learned reward model could faithfully represent human preferences, as our rating rubric considers the visual overlap as the most important factor that determines prototype quality.

# 6 GENERALIZABILITY OF THE R3 FRAMEWORK

In addition to ProtoPNet, we also tested the generalizability of our R3 framework on ProtoPFormer (Xue et al., 2022), which is another recent ProtoPNet extension that uses vision transformers (ViTs) backbones. With the exact same debugging procedure, similar performance gains in interpretability and accuracy are also observed, and the detailed experiment results can be found in Appendix E.

# 7 LIMITATIONS AND FUTURE WORK

While R3-ProtoPNet succeeds in bringing forth interpretability and predictive performance gains, there's still room for improvement. For example, the reward model is trained on ratings of a single image and heatmap, highly constrained to measuring overlap between prototype and the object of interest while ignoring cross-image preferences, such as body-part consistency. We also note that R3-ProtoPNet fails to entirely eliminate duplicates, with several high-reward prototypes converge to the same part of the image. To address these issues, it's promising to extend ratings to multiple images and heatmaps and create more diverse reward models, possibly using them in ensemble.

Another limitation with R3-ProtoPNet and other methods that rely on human feedback is that the model itself might be learning features that, while seemingly confusing to a human, are helpful and meaningful for prediction. Barnett et al. (2021) argue that the ProtoPNet can predict with non-obvious features like texture and contrast, which might be penalized via a learned reward function. An interesting line of future work is to investigate how certain ProtoPNet variants could critique human feedback, and argue against a human-biased reward function.

While this work focuses on increasing the performance of ProtoPNet, a major benefit of reward-based finetuning is its flexibility in application. With proper adaptations, we expect our R3 debugging framework could generalize to many other prototype-based or interpretable machine learning models and serve as a useful concept-level debugging tool.

# 8 CONCLUSION

In this work, we propose the R3-ProtoPNet, a method that uses a learned reward model of human feedback to improve the meaningfulness of learned prototypical parts. We find that ensembling multiple R3-ProtoPNets results in increased performance over original ProtoPNet ensembles. Considering the high performance of the reward model, we use the reward model as a measure of prototype quality alongside the established measure of activation precision, allowing us to critique the interpretability of ProtoPNet along a human lens. As demonstrated here, the ability of reward learning to quantify qualitative human preferences make reward-based fine-tuning a promising direction for the improvement of interpretable deep models.

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

## A  ALGORITHM DESCRIPTION

---

**Algorithm 1** Reward Reweighed, Reselected, and Retrained Prototypical Part Network (R3-ProtoPNet)

---

1: **Initialize:** Collect high-quality human feedback data and train a reward model.
2: **Reward Reweighing:** Perform the reward-reweighted update for the ProtoPNet, defined as follows:

$$\max_{p_j} \mathcal{L}_{reweigh}(z_i^*, p_j) = \max_{p_j} \sum_{i \in I(p_j)}^{n} r(x_i, p_j) \frac{1}{\lambda_{dist}\|z_i^* - p_j\|_2^2 + 1}$$

Optimize this loss function, which leads to locally maximal solutions, improving the prototypes.
3: **Prototype Reselection:** Run the reselection procedure based on a reward threshold. If $\frac{1}{n_k} \sum_{i \in I(p_j)} r(x_i, p_j) < \alpha$, reselect the prototype by sampling from patch candidates and temporarily setting the prototype to a new candidate that passes the acceptance threshold and is unique from other current prototypes.
4: **Retraining:** Retrain the model with the same loss function used in the original ProtoPNet update, to realign the prototypes and the rest of the model.

---

We provide a high-level description of R3-ProtoPNet in Figure 2. Reading from top to bottom, it illustrates the initial ProtoPNet training step, followed by reward modelling, and then the R3 updates. We also provide a high-level summary that describes the complete R3 training in Algorithm 1. We note that $z_i^* = \operatorname{argmin}_{z \in \text{patches}(f(x_i))} \|z - p_j\|_2^2$, $I(p_j) = \{i \mid y_i \in \text{class}(p_j)\}$, $\lambda_{dist}, \alpha$ are predetermined hyperparameters, and $n_k = |I(p_j)|$, the number of training images in class $k$.

We trained ProtoPNet and R3-ProtoPNet on a single A100 GPU. The ProtoPNet training took about 10 hours and required at most 100 epochs. The R2 update for each prototype needed at most 100 iterations and it took roughly 90 minutes to complete an R2 update for a ProtoPNet with 10 prototypes per class. The final retraining step for R3-ProtoPNet required another 20 epochs and lasted roughly 3 hours.

## B  PAIRWISE LOSS FUNCTION FOR THE REWARD MODEL

For completeness, here is the explicit formulation of the loss function described in Section 4.2:

$$\mathcal{L}_{reward} = -\sum_{i \neq i' \text{ or } j \neq j'} \left[ \mathbf{1}_{c_{i,j,i',j'}=-1} log(\frac{exp(r(x_i, h_{i,j}))}{exp(r(x_i, h_{i,j})) + exp(r(x_{i'}, h_{i',j'}))}) \right.$$

$$\left. + \mathbf{1}_{c_{i,j,i',j'}=1} log(\frac{exp(r(x_{i'}, h_{i',j'}))}{exp(r(x_i, h_{i,j})) + exp(r(x_{i'}, h_{i',j'}))}) \right]$$

where $c_{i,j,i',j'}$ refers to the comparison value associated with the column indexed by $i, j, i', j'$ in the synthetic dataset $\mathcal{D}_{paired}$, which is explained in section 4.2. The architecture of the reward model is detailed in section 5.2.

## C  THRESHOLDS AND THE NUMBER OF UPDATED PROTOTYPES

As described in Section 4.3, for each base architecture, the various thresholds for reweighing, reselection, and acceptance. These thresholds were chosen by examining the reward distribution of the base architectures to see if prototypes with low reward cluster around any particular values. Across models, a reweighing threshold of $0.4$ or $0.35$ sufficed, but further tuning was needed for the reselection and acceptance thresholds. We present the final thresholds used for each R2 step in Table 4.

Using the reselection thresholds above, we report the total number of updated prototypes for each base architecture in Table 5.

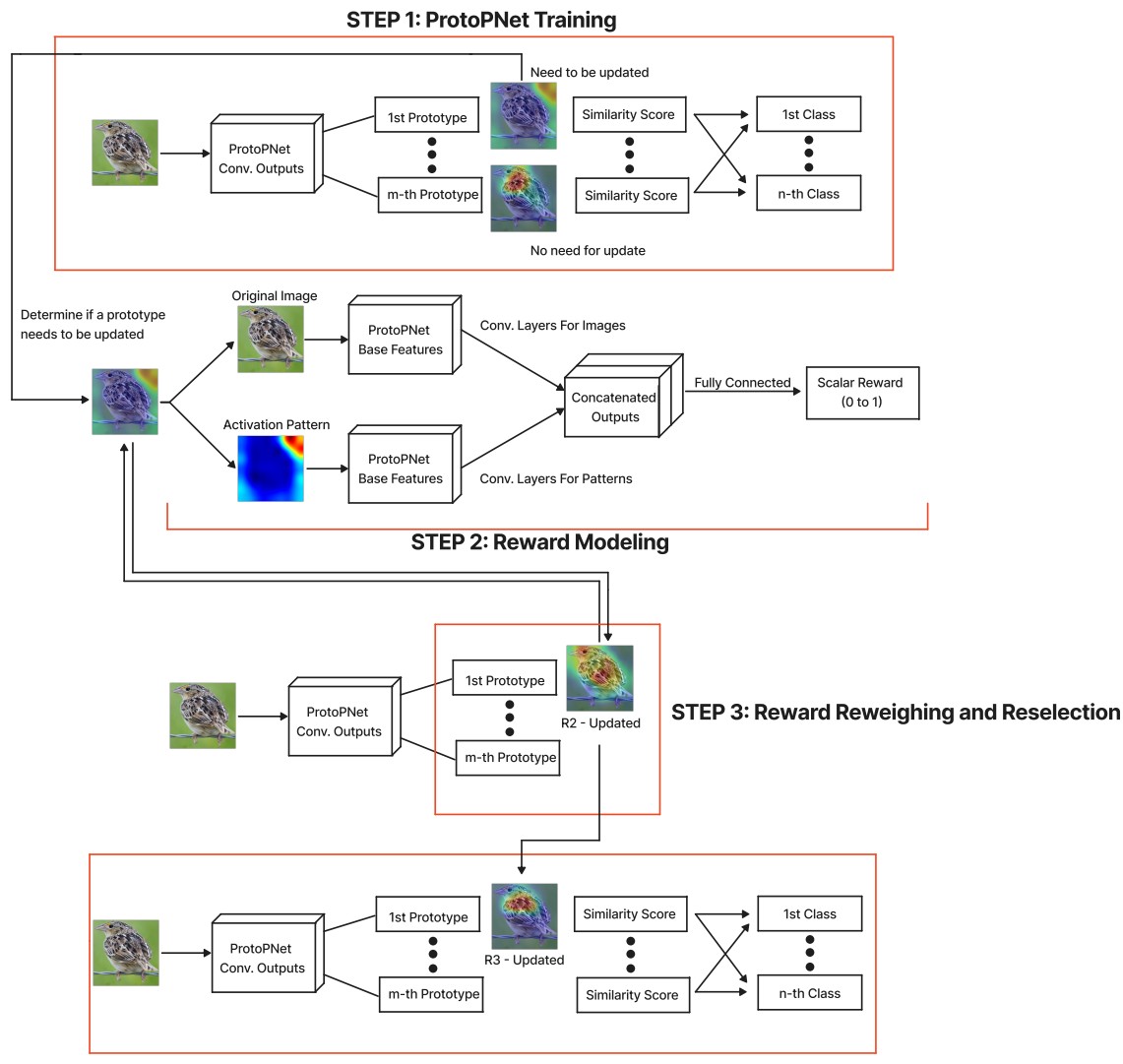

Figure 2: A step-by-step illustration of R3-ProtoPNet.

| Base ($m_k$) | Reselection Threshold | Reweigh Threshold | Acceptance Threshold |
|---|---|---|---|
| VGG-19 (5) | 0.35 | 0.40 | 0.50 |
| VGG-19 (10) | 0.25 | 0.40 | 0.43 |
| ResNet-34 (10) | 0.22 | 0.35 | 0.40 |
| ResNet-50 (10) | 0.18 | 0.35 | 0.40 |
| DenseNet-121 (10) | 0.25 | 0.35 | 0.45 |
| DenseNet-161 (10) | 0.25 | 0.35 | 0.43 |

Table 4: Thresholds used across base architectures during R2 step.

| Base ($m_k$) | Reselected Prototypes | Reward-Reweighed Prototypes |
|---|---|---|
| VGG-19 (5) | 107 / 1000 | 231 / 1000 |
| VGG-19 (10) | 224 / 2000 | 504 / 2000 |
| ResNet-34 (10) | 189 / 2000 | 622 / 2000 |
| ResNet-50 (10) | 205 / 2000 | 549 / 2000 |
| DenseNet-121 (10) | 334 / 2000 | 462 / 2000 |
| DenseNet-161 (10) | 176 / 2000 | 458 / 2000 |

Table 5: Total number of prototypes updated per base architecture across the two R2 steps, divided by the total number of prototypes for that network.

## D    SENSITIVITY TO AMOUNT OF HUMAN FEEDBACK

To evaluate the influence of the amount of human feedback on our R3 framework, we tried using fewer human ratings to train a reward model and then perform the R3 updates. The results are in Table 6. Although we used 500 ratings to reach the peak performance in the main experiments, it's observed that the R3 framework is able to improve the original ProtoPNet even when there are only 300 ratings.

| Metric \ #Ratings | 100 ratings | 200 ratings | 300 ratings | 400 ratings | 500 ratings |
|---|---|---|---|---|---|
| Reward Model Acc. | $69.50 \pm 0.25$ | $77.34 \pm 0.25$ | $83.27 \pm 0.27$ | $88.67 \pm 0.16$ | $90.09 \pm 0.20$ |
| R3-ProtoPNet Acc. | $77.64 \pm 0.34$ | $80.35 \pm 0.24$ | $81.42 \pm 0.29$ | $82.60 \pm 0.22$ | $82.86 \pm 0.16$ |

Table 6: Test accuracies of the trained reward models and the resulting R3-ProtoPNets (ensembled) given different amount of human ratings.

## E    PERFORMANCE OF R3-PROTOPFORMER

To test the generalizability of our R3 framework, we applied our R3 framework to ProtoPFormer (Xue et al., 2022), which is another ProtoPNet extension that uses vision transformer (ViT) backbones. In ProtoPFormer, two types of prototypes are used: the global prototypes are able to provide holistic views of the objects and eliminate confounding effects of the background, while the local prototypes capture the fine-grained visual features that are useful for classification. Empirically we found success applying our R3 framework to update both global and local prototypes. The results are summarized in Table 7.

| Metric | ProtoPFormer | Reselected | Reweighed | R3-ProtoPFormer |
|---|---|---|---|---|
| Test Accuracy | $82.27 \pm 0.15$ | $73.35 \pm 1.53$ | $71.79 \pm 2.08$ | $84.31 \pm 0.21$ |
| Avg. Reward (global) | 0.40 | 0.45 | 0.48 | 0.49 |
| Avg. Reward (local) | 0.46 | 0.52 | 0.55 | 0.53 |
| Activation Precision (global) | $33.48 \pm 0.22$ | $36.59 \pm 0.96$ | $42.96 \pm 1.56$ | $45.43 \pm 1.61$ |
| Activation Precision (local) | $47.63 \pm 0.38$ | $50.28 \pm 1.31$ | $53.11 \pm 1.72$ | $53.36 \pm 1.24$ |

Table 7: Performance report of R3-ProtoPFormer across different stages. We used the best-performing DeiT-S backbone.

# F    PROTOTYPE EXAMPLES

Here we provide some examples of prototypes from ProtoPNet, R2-ProtoPNet, and R3-ProtoPNet. In Figure 3, we plot the closest image patch of the 5 prototypes trained on the VGG-19 base architecture for four randomly selected classes from the CUB-200-2011 dataset. The first row in each of blocks of images consists of the prototypes from ProtoPNet, the second row consists of prototypes from R2-ProtoPNet, and the third row consists of prototypes from R3-ProtoPNet. We see that the R2 update indeed centers otherwise off prototypes on the bird in question, and the complete R3 update makes those prototypes helpful for prediction.

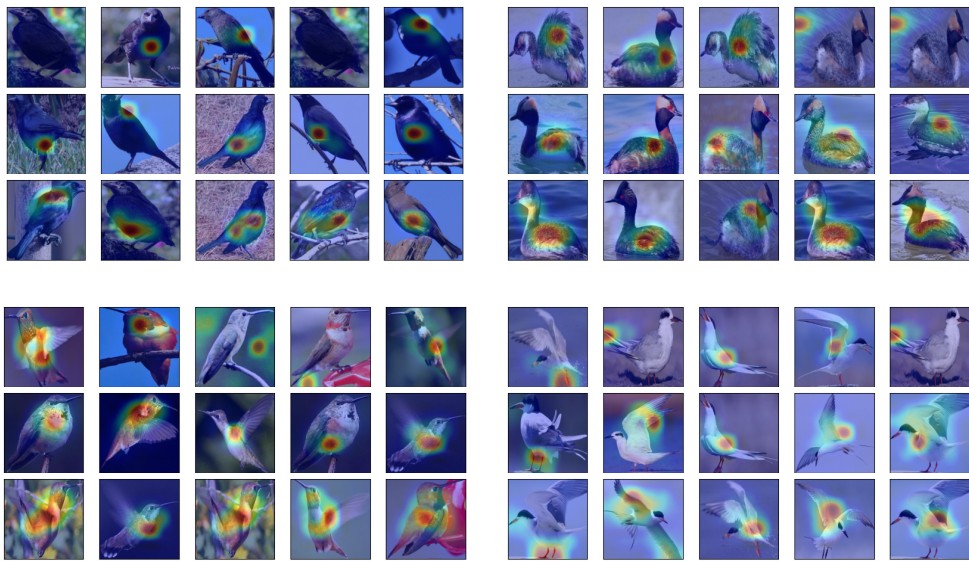

Figure 3: Closest training patches of the five prototypes of ProtoPNet (top row), R2-ProtoPNet (middle row), and R3-ProtoPNet (bottom row) within the same class (5 prototypes per class). Each cluster of 3 rows of images is a seperate class.

In Figure 4, we visualize one prototype (column) across the same image. This Figure illustrates how the prototype changes during the R2 and R3 update when the image is held fixed. We see that prototypes are indeed localized on the birds, without having dependence on spurious features like the background.

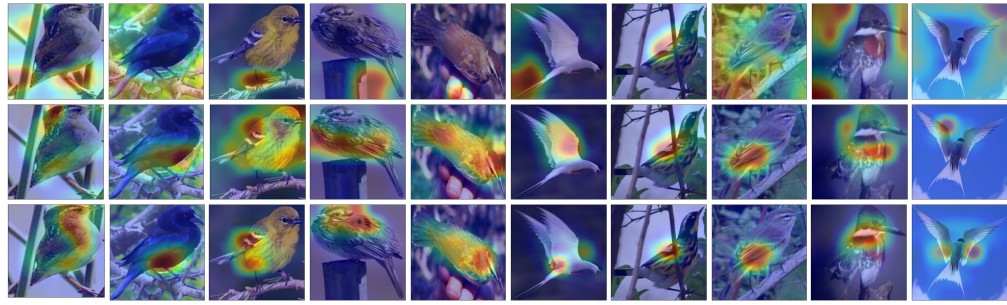

Figure 4: Prototype projections on the same image (each column) from ProtoPNet (top row), R2-ProtoPNet (middle row), and R3-ProtoPNet (bottom row).

