# OpenReview forum: "Improving Prototypical Part Networks with Reward Reweighing, Reselection, and Retraining"
_ICLR.cc/2024/Conference — Submitted to ICLR 2024_

### Official Review · Reviewer_KfuC · 2023-10-31

**Soundness:** 3 good
**Presentation:** 3 good
**Contribution:** 2 fair
**Rating:** 6
**Confidence:** 4

**Summary:**

The paper proposes a new method called R3-ProtoPNet to improve the interpretability and performance of the prototypical part network (ProtoPNet). ProtoPNet is an interpretable image classifier that makes predictions based on prototypical parts of images. However, it can learn spurious or inconsistent prototypes. R3-ProtoPNet collects human feedback on prototype quality to train a reward model. The reward model predicts human preferences between prototypes. R3-ProtoPNet has 3 stages - reward reweighting, prototype reselection, and retraining. Reweighting uses the reward model to update prototypes to be more similar to highly rewarded patches according to human preference. Reselection replaces low reward prototypes with random high reward patch candidates. Retraining realigns the network features and classification layer with the improved prototypes. Experiments were done on CUB-200-2011 birds dataset using VGG, ResNet, and DenseNet architectures. Human ratings of prototype-image pairs were collected via Amazon Mechanical Turk.
The reward model achieved over 90% accuracy in predicting human prototype preferences. R3-ProtoPNet improved average prototype reward by 31.65% and activation precision by 46.85% over ProtoPNet.

**Strengths:**

Advantages of the approach:
Inspired by the RLHF, the authors use a flexible human feedback mechanism via learned reward model. Reward model also provides a quantifiable metric for prototype quality. Reweighting and reselection improve prototype interpretability. Retraining maintains or improves predictive performance.
This approach can be applied to different base architectures like VGG, ResNet, etc.
Most importantly, reward-guided training aligns prototypes to human preferences.
The results are quite compelling: it achieves high reward model accuracy in predicting human preferences -- this indicates it captures prototype quality well. Increased average reward shows prototypes are more meaningful after R3 training.
Improved activation precision verifies R3 prototypes have better overlap with birds. R3 training maintains or improves accuracy, showing no loss of predictive power. R3 ensemble outperforms ProtoPNet ensemble, demonstrating improved performance.
Examples show reduced dependence on background and other spurious features after R3 training.
Thus, retraining enables improved prototypes to be utilized for better prediction.

**Weaknesses:**

Some potential weaknesses include:
The approach strikes as overly hand-crafted. The beauty of ProtoPNet was exactly in that it learned parts from coarse labels. Compared to that, the proposed approach is overly reliant on guidance by manually collected labels.
Limited evaluation on just one dataset (CUB birds) - needs more diverse evaluation.
Duplicate prototypes still occur after R3 training.
Reward model may not sufficiently capture cross-image consistency.
Potential for reward model to penalize useful but non-obvious features.

**Questions:**

You showed results on the CUB birds dataset. How does R3-ProtoPNet perform on more complex, diverse datasets like ImageNet? Does it still improve interpretability and accuracy?
Can you provide some analysis or examples showing that R3 training does not lose focus on useful non-obvious features like texture or contrast?
How sensitive is R3-ProtoPNet to the amount and quality of human feedback data for the reward model? How little data can you use and still see benefits?
You retrain the full network - have you tried only retraining the classifier layers? This could improve efficiency.
How does R3-ProtoPNet scale to larger datasets and models? Is the computational overhead of R3 prohibitive?

---

> ### Author Response · Authors · 2023-11-19
> **Author Rebuttal #4**
>
> Thanks for your constructive feedback, and we will try to improve the paper as much as we can based on your suggestions (please check out our updated version, which will be uploaded soon). We will address your main concerns and questions below:
>
> $\textbf{The method seems overly hand-crafted:}$
> We agree with the reviewer about the beauty of ProtoPNet being able to learn parts from course labels, and we believe that R3-ProtoPNet is actually an extension of that simplicity. As discussed in our work, the original ProtoPNet fails appropriately learn consistent and meaningful prototypes, but does manage to learn from parts of the images. If one wished to resolve these issues without manually collected labels, it would still be necessary for the designer of the system to enforce a bias against spurious and inonsistent prototypes via a penalty in the loss function, or, in the case of the original ProtoPNet paper, designing pruning methods to throw out spurious information.
>
> In our opinion, the beauty of the R3-ProtoPNet is precisely that it does not require such a penalty, but instead relies on the quick collection of human feedback.
>
> $\textbf{More diverse evaluation on other datasets:}$
> Thanks for pointing this out! We chose the CUB bird dataset because it's the most commonly used dataset in related works and identifying bird species is a task that could elicit very natural human preferences. We are currently working on evaluating our R3 framework on other datasets such as the Stanford Cars dataset. We expect similar performance gain, but the full results on all base architectures might not come out by the end of the discussion period. We will definitely include those results later.
>
> $\textbf{Generalization to other models:}$
> We have tested our method on ProtoPFormer, which is a transformer-based ProtoPNet (Xue et al. 2022), and we've shown that our R3 framework also leads to better interpretability and improved accuracy. The results will be reported in our modified version of the paper. Furthermore, the high-level notion of our R3 framework could be thought of as a type of RLHF that acts on the interpretability of a given model. We believe even for a model without prototypes, as long as it has an observable intermediate stage that's possible to use a reward model to predict human preferences and thus to evaluate its interpretability, with some modifications our R3 framework could be applicable.
>
> $\textbf{The possibility of losing non-obvious but useful such as texture and contrast:}$
> Whether this could happen really depends on the learned reward model. In our rating rubric for the bird dataset, the prototype will have a score of at least 4 as long as a substantial part of it is overlapped with the bird body, so it's not likely for a prototype capturing non-obvious features on the body to be updated significantly and lose those features completely - we included plenty of visualizations in the appendix. On the other hand, since the goal of our R3 framework is to make the model more interpretable and align with the preferences of the users, introducing human biases to the classification problem is expected and choosing whether to believe them is more of a philosophical question rather than an issue with our R3 framework.
>
> $\textbf{How sensitive is R3-ProtoPNet to the amount and quality of human feedback data for the reward model:}$
> Actually we had a recent empirical analysis of the relationship between the amount of human ratings and the predictive performance of R3-ProtoPNet and the reward model. We will include this in our modified paper.
>
> $\textbf{Only retraining the classifier layers:}$
> Empirically we found that only retraining the classifier layers would lead to slightly worse performance than retraining the entire ProtoPNet. This is because the last layers operate on the similarities between the convolution outputs of the base model and the prototypes: once we update our prototypes (especially after the reward reweighing step), we'd also need to update the base features slightly to allow for better alignment.
>
> $\textbf{Computation overhead of R3:}$
> The computation cost of R3 is discussed in Appendix A, and we think several hours of extra debugging/training time is acceptable. We'd also like to note that the users of R3-ProtoPNet could heuristically control the proportion of bad prototypes to be updated by changing the reselection and reweighing thresholds (Appendix C) and thus control the runtime.

---

### Official Review · Reviewer_Tqnf · 2023-10-31

**Soundness:** 3 good
**Presentation:** 2 fair
**Contribution:** 2 fair
**Rating:** 6
**Confidence:** 3

**Summary:**

In this paper, the authors proposed a reward reweighed, reselected, and retrained prototypical part network (R3-ProtoPNet) that improves upon the original prototypical part network (ProtoPNet) by Chen et al. (2019). Given a trained ProtoPNet, the proposed R3-ProtoPNet involves: (1) collecting human feedback regarding the quality of learned prototypes; (2) learning a reward function that takes as input an image and its prototype activation map from a particular prototype, and outputs a reward that represents how good the prototype activation map is for the given image; (3) using the learned reward function to reweigh the inverse distance to a prototype and to improve the prototype; (4) reselecting prototypes whose rewards are below a threshold by replacing them with random patch representations whose rewards are above the acceptance threshold; (5) retraining the model to improve the prediction accuracy. The authors performed extensive experiments using several ProtoPNet models with various base architectures, and concluded that their R3-ProtoPNet has higher test accuracy, higher average rewards (based on the learned reward function), and higher activation precision over the original ProtoPNet.

**Strengths:**

- The authors proposed a way to improve the prediction accuracy of ProtoPNet, which is an important interpretable deep classifier.
- The proposed method is generally sound.
- The paper is easy to read.

**Weaknesses:**

- There is no evaluation of the learned reward function.
- While the proposed R3-ProtoPNet does empirically improve the prediction accuracy over the original ProtoPNet, there is little theoretical insight as to what makes R3-ProtoPNet work.
- Is it necessary to have both reward reweighing and prototype reselection? More specifically, while prototype reselection is more necessary (it is a way to move away from badly learned/low-reward prototypes), reward reweighing seems optional to me. An ablation study on this will be helpful.
- There are limited visualizations. While the authors did include visualizations of prototypes (in the appendix), they did not include enough examples of how prototypes are used in R3-ProtoPNet, the closest prototypes to a given image from a trained R3-ProtoPNet, and the closest image patches to a given prototype learned by an R3-ProtoPNet. These are needed to convince readers that an R3-ProtoPNet uses high-quality prototypes in reasoning, and the learned prototypes are semantically meaningful.

**Questions:**

- An R3-ProtoPNet can be thought of as initializing a ProtoPNet in a smart way (by reward reweighing and prototype reselection), and then training it again. As mentioned earlier, is there any explanation for why this empirically improves the prediction performance?
- As mentioned before, is it necessary to have both reward reweighing and prototype reselection? Please include an ablation study on this.

**Details Of Ethics Concerns:**

N/A.

---

> ### Author Response · Authors · 2023-11-19
> **Author Rebuttal #3**
>
> Thanks for your constructive feedback, and we will try to improve the paper as much as we can based on your suggestions (please check out our updated version, which will be uploaded soon). We will address your main concerns and questions below:
>
> $\textbf{No evaluation of the learned reward function:}$
> We mentioned in 5.2 that our reward model achieves 90.09\% test accuracy on the synthetic dataset (the held-out test set has about 13k pairs of heatmaps), which means it could effectively predict the relative order of prototype quality and thus model the collected human preferences reliably.
>
> $\textbf{Why our method empirically improves the performance:}$
> In the original ProtoPNet, the prototypes are generated by the model itself only based on supervised predictive signal, and the interpretable reasoning chain is maintained by the "push" operation, which sets the prototypes to be the closest input image patch. However, one issue with this process is that once initialized the prototypes cannot change dramatically, so the bad-initialized (possibly due to some predictive shortcuts) ones will always remain bad - we consider this as the main cause of the spurious features in the ProtoPNet. In our R3-ProtoPNet, with the help of an external reward model, reselection allows the global movement/reinitialization of the prototypes, while reweighing allows local movement of the prototypes to be more aligned with human preferences (e.g. more overlap, more characteristic features, etc.).
>
> $\textbf{Reward reweighing seems optional:}$
> As stated above, reweighing allows local movement of the prototypes to be more aligned with human preferences (e.g. moving an originally partially overlapped prototype more toward the body), and we did find both interpretability and accruacy gains as the result of reward reweighing. Please refer to 1) our updated performance report tables which will break the R2 step into reselection and reward reweighing as well as 2) an ablation study added in the appendix. Thanks for the suggestion!
>
> $\textbf{More illustrative visualizations:}$
> Actually We included the closest image patches to each prototype and the prototypes corresponding to each image during different stages in the appendix. That being said, we will consider adding more illustrative visualizations in our modified paper and also revising the captions, and thanks for the feedback.

---

### Official Review · Reviewer_cp6j · 2023-11-01

**Soundness:** 2 fair
**Presentation:** 3 good
**Contribution:** 3 good
**Rating:** 5
**Confidence:** 4

**Summary:**

The authors propose a debugging procedure for Part-prototype Networks based on human feedback.  Rather than using the feedback as-is, as done by prior work, the proposed method (R3-PPNet) uses it to train a reward function, which generalizes said feedback, and use the latter to drive the model away from bad prototypes.  The model refinement step uses a couple of heuristics to improve the model.  Experiments are carried out on the CUB-200 dataset augmented with prototype rating feedback collected with Mechanical Turk.

**Strengths:**

**Originality**: To the best of my knowledge, the idea of combining RLHF mechanics and concept debugging (also, I like it a lot).

**Quality*:  The proposed technique is sensible (all three stages), and makes good use of existing techniques.  The coverage of related work is good.  The experimental setup is mostly satisfactory (for instance, the authors consider several backbones and good evaluation metrics), but see below.  I also appreciate how the authors were upfront about limitations of their technique.

**Clarity**: The text is very readable.  Ideas are conveyed clearly.

**Significance**:  This work tackles an important problem in concept-based models.  The key contribution is, in my opinion, showing that RLHF can be used for debugging learned concepts (or steering the model towards using better concepts).  The specific algorithm itself is not core, and it could be improved, I think.  Regardless, I believe the main contribution will have some impact on interactive debugging techniques.

**Weaknesses:**

**Originality**:  This work combines existing ideas.  The degree of novelty, from a technical perspective, is limited (but still sufficient, in my opinion).

**Quality*:  [Q1] One issue with the experiments is that they consider a single data set (CUB-200).  Bontempelli et al. (cited by the authors) do evaluate their approach on three data sets (CUB-200, a synthetic data set, and an X-ray data set).  The choice of focusing on CUB-200 only is not exactly justified.

[Q2] It is also not clear why R3-PPNet was not compared to the work of Bontempelli et al. -- it should be trivial to convert tratings into binary lables (say, ratings below 3 could be converted to a "bad" label, and 4 and above to "good").

[Q3] One clear downside of the approach is that the reward model is pre-trained on a large number (in terms of annotation cost) of ratings.  While the cost of collecting ratings is generally compensated for when dealing with LLMs (as these models can be used for a variety of tasks, so you'd want them to be as good as possible), the cost-benefit ration for ProtoPNets is not as clear.  I think this should, at the bare minimum, be discussed in the limitations.


**Minor issues**:

- In Section 5.1, you wrote "R3-ProtoPNet requires two datasets", but that's not true.  It needs one dataset with additional annotations:  now new *inputs* are added by this second "dataset".  I'd prefer if the text was changed accordingly.

**Questions:**

Please see Q1-Q3 above.

Q4.  It seems to me R3-PPNet is designed for passive learning only.  The reason is that, if I understand correctly, the reward function (which depends on the learned model) becomes obsolete after fine-tuning/debugging the model.  If the debugged model is still buggy, the old reward function cannot be used again.  Is this correct?

Q5.  How have alpha, beta, and gamma been chosen in the experiments?  How should users of your system choose them?  How sensitive is the quality of the resulting model to the choice of thresholds?

---

> ### Author Response · Authors · 2023-11-18
> **Author Rebuttal #2**
>
> Thanks for your constructive feedback, and we will try to improve the paper as much as we can based on your suggestions (please check out our updated version, which will be uploaded soon). We will address your main concerns and questions below:
>
> $\textbf{Only one single bird dataset is used: }$
> Thanks for pointing this out! We chose the CUB bird dataset because it's the most commonly used dataset in related works and it could elicit very natural human preferences. We are currently working on evaluating our R3 framework on other datasets such as the Stanford Cars dataset. We expect similar performance gain, but the full results on all base architectures might not come out by the end of the discussion period. We will definitely include those results later.
>
> $\textbf{Why R3-PPNet was not compared to the work of Bontempelli et al.: }$
> One major distinction we'd like to point out is that R3-ProtoPNet is an offline debugging framework: it only requires one batch of human feedback to train the reward model, and this happens before we actually debug the ProtoPNet. On the other hand, the ProtoDebug framework proposed by Bontempelli et al. requires iterative online human feedback on the exact models and prototypes to be updated. Furthermore, to the best of our knowledge, ProtoDebug is not designed for large-scale prototype debugging: the required human labor would scale up with the number of prototypes (their collected feedback and evaluation was on 5 out of 200 classes while we evaluated on all 200 classes); by contrast our method only needs a fixed number of human annotations to train the reward model. Given the reasons above, we don't think the two frameworks are quite comparable.
>
> $\textbf{Cost-benefit analysis of collecting human labels: }$
> While a large and high-quality dataset of human feedback is required for RLHF to improve large language models and other complex generative models, work on preference learning in RL demonstrated that relatively small amounts of human feedback was necessary to learn a useful reward model, as long as the data to be rated is chosen carefully (Christiano et al. 2017). Generally, it is our understanding that the amount of human feedback data needed to learn a particular reward model is dependent on the underlying complexity of the feedback task. And, for learning reward functions relevant to ProtoPNet, the feedback task does not require large amounts of human feedback data.
>
> For our method of learning our reward function, we empirically found that ~500 rated heatmaps would be able to train a reward model with over 90\% accuracy (it would converge afterward), and although we collected multiple user ratings for these samples using Mechanical Turk and took the average, we found that it took roughly one hour for a single human rater to provide high-quality ratings for all 500 samples. As stated earlier, since the trained reward model could be reused for different base architectures and prototypes, we don't think the labor cost outweighs the benefits.
>
> $\textbf{The reward function becomes obsolete after fine-tuning/debugging the model: }$
> Yes, this is largely correct. And if some prototypes are still buggy after the R3 update, it's most likely that it's not detected/properly corrected by the learned reward model, and such cases do happen due to the noise in the entire human-in-the-loop module. Conventional ways to robustify the reward model include comprehensive selection of the heatmaps (so that most typical errors are included) to be rated and increasing the number of raters (which we did). One interesting future direction would be having a ProtoPNet updated by multiple learned reward models sequentially, each capturing one single aspect of human preference (e.g. overlap, body part consistency, etc.), so that each reward model would be less noisy because of the single learning objective.
>
> $\textbf{Alpha, Beta, and Gamma: }$
> As discussed in section C of the appendix, empirically those hyperparameters are determined by comparing the reward distribution of the prototypes to the visualizations of the heatmaps: first the reselection threshold should separate those completely-off prototypes from the rest, and then the reweighing threshold should separate the prototypes that still need local updates from those already look good. Finally we want to set the acceptance threshold such that it's large enough to improve the quality of the reselected prototypes while not causing too many reselection failures. We found that given a properly annotated synthetic dataset, fixed thresholds would lead to approximately the same performance for the same base architecture, and bad thresholds would influence the number of updated prototypes in the R2 step and make the R3-ProtoPNet not reaching peak performance (but usually no worse than the original model).

---

> > ### Comment · Reviewer_cp6j · 2023-11-23
> > **Reply to authors**
> >
> > Thank you for your detailed reply.
> >
> > **Single dataset**: I agree CUB is a good choice, but it is also true that claims made on a single data set may not generalize easily to others (see below).  This is a clear weakness of the current manuscript and (understandably) it cannot be solved in a timely fashion.  This is unfortunate.
> >
> > > One major distinction we'd like to point out is that R3-ProtoPNet is an offline debugging framework
> >
> > Agreed.  I believe this distinction should be clarified in the text.
> >
> > > ProtoDebug is not designed for large-scale prototype debugging: the required human labor would scale up with the number of prototypes (their collected feedback and evaluation was on 5 out of 200 classes while we evaluated on all 200 classes); by contrast our method only needs a fixed number of human annotations to train the reward model.
> >
> > It is possible that R3 achieves better results with less supervision, but this kind of claim should be verified empirically.
> >
> > > While a large and high-quality dataset of human feedback is required for RLHF to improve large language models and other complex generative models, work on preference learning in RL demonstrated that relatively small amounts of human feedback was necessary to learn a useful reward model, as long as the data to be rated is chosen carefully (Christiano et al. 2017).
> >
> > I can agree with this.
> >
> > > And, for learning reward functions relevant to ProtoPNet, the feedback task does not require large amounts of human feedback data.
> >
> > In the context of CUB200.  Collecting feedback on X-ray scans would be more expensive (as it requires dealing with experts, which cannot be normally found on MTurk). Testing on more data sets would be useful to gauge this assumption.
> >
> > > One interesting future direction would be having a ProtoPNet updated by multiple learned reward models sequentially
> >
> > This is an interesting idea!
> >
> > All in all, your rebuttal helped me to better understand some points.  Still, the fact that the analysis is restricted to a single data set is a clear weakness.  I will decide whether to increase the score after discussing this with the other reviewers.

---

### Official Review · Reviewer_Fuan · 2023-11-10

**Soundness:** 2 fair
**Presentation:** 3 good
**Contribution:** 2 fair
**Rating:** 6
**Confidence:** 4

**Summary:**

The authors propose three modifications to the training of prototypical part networks, inspired by RLHF (i..e using lay human feedback on the quality of prototypes). The authors demonstrate that this approach improves the interpretability of the learned prototypes according to three metrics, and provide qualitative support for the improvements conferred by their approach.

**Strengths:**

Clear relationship with differentiation from/discussion of improvements from protopnet.
Overall well written, specially the section on 4.2. Very good integration of notes, explanation, and notation/math (this is surprisingly rare, so good job!).

**Weaknesses:**

While it's great that your scope is clear, it ends up feeling a bit like a lab report, where your assignment was to apply RLHF to ProtoPNet and report the results. This isn't bad per-se; science for the sake of doing it can be great, but ideally for a scholarly paper I would want to see some more interpretation, insight, and high-level thinking.

It would also be nice to see some more illustrative/selective qualitative results.

Finally, it would be ideal to have an evaluation done by expert human evaluators (whatever that group might be depending on the intended goal of the paper, if any) -- ie. most likely, ornithologists or birders. Given that the focus is on interpretability, how useful are the prototypes actually, to humans who might use them? Do they find the same characteristic features people use for ID? Any novel and interesting ones?

Captions of tables could be improved to be more descriptive.

**Questions:**

The "parts" highlighted by the original or your modified version tend to be very soft-edged and blobby. In some sense I would "like" to see prototypes that crisply highlight e.g. tails, feet, beaks, etc. Do you think this is desirable and/or possible? Why or why not? E.g. the pixel-level segmentation maps used in AP seem like they could provide very good supervisory signal for this.
I also feel there could be a more nuanced discussion of what makes a feature spurious or not. E.g. it's repeatedly stated that the background is spurious, but (as an amateur birder) I can tell you habitat (especially if it includes something as characteristic as a nest) can often be incredibly useful for identification. Even the presence of sky vs. grass or sea or something could be informative (e.g. woodland thrushes would virtually never be photographed against open sky, unlike a swallow). And even at a more general level, the shape/contour of e.g. tailfeathers or the bird's overall silhouette (i.e. the edge pixels which would include some bg) are also often characteristic.
The main stated high-level goal of the work is to improve interpretability ... for whom, and for what purpose? If everything worked "perfectly", what would this system be able to do/be used for? Or if it doesn't have a goal in mind (which again, IMO is totally fine and great for science), what do we understand better about RLHF or Prototype nets or deep nets in general as a result of your work?

small things:- protopnet should be cited in abs- briefly state what results/evidence you found supports the claims about improvement (second to last sentence) - suggest renaming the section "limitations" to "limitations of ProtoPNet" to avoid confusion with typical "limitations" sections. - interpretability ...leads to a .... suggest rephrasing as "interpretabilty ... is useful for RLHF" or something like that.  - explain what is the Bradley Terry Model and why you want to use it

---

> ### Author Response · Authors · 2023-11-18
> **Author Rebuttal #1**
>
> $\textbf{The paper has limited scope:}$
> While we agree that the paper is focused on applying RLHF to ProtoPNet, we respectfully disagree with the reviewer that the paper has limited scope. Many interpretability or explainability methods in deep learning are designed for a particular architecture, data modality, or problem in mind. Part of the difficulty of interpretable deep learning is designing or utilizing interpretable methods for particular problems. This process usually involves intense and time-consuming exploration of the data and understanding of the problem. Combining RLHF with existing interpretability methods, while certainly not eliminating the need for a deep understanding of the problem at hand, has the potential to simplify the modeling process, allowing researchers and practitioners to spend less time designing and building complex methods, and more time interrogating the reasoning and limits of their models. We believe that R3-ProtoPNet is one such example of how RLHF and adjacent ideas can allow for improvements to interpretable deep learning more generally.
>
> An example of this is that ProtoPNet fails to provide consistent prototype labels across images, where a single prototype can still map to multiple parts of a bird across images. Instead of designing a hand-crafted regularization function to promote cross-image consistency, we can now instead consider how to best elicit and model human feedback on cross-image consistency of prototypes, which is something human viewers can do with ease.
>
> We will revise our paper to include a discussion of the general contribution to RLHF and interpretability.
>
> $\textbf{Human expert evaluation needed:}$
> The CUB-200-2011 dataset is a commonly used benchmark for evaluating ProtoPNets due to the ease with which an amateur can recognize how a method makes classifies a bird, such as focusing on the background or the bird itself, on the bird's wings or head--or some uninterpretable combination of the two. We agree with the reviewer that the utility of such a bird classification method to an actual expert is unclear, given that no expert evaluation is used to justify this method. We hope to provide such an exploration in future work.
>
> $\textbf{Prototypes are sometimes soft-edged and blobby:}$
> In ProtoPNet, a heatmap is generated by calculating the similarity of the upsampled prototype and each of the 49 patches (each of size 32x32) of an original image, so if the textures of several neighboring patches are similar, then the highly activated region would inevitably be larger/blobby. In fact we would consider this as more of a visualization problem. As we can see in the visualized examples in the appendix, many prototypes are able to clearly focus on wings, tails, etc. We hypothesize that some prototypes don't focus on specific body parts because sometimes more subtle visual information such as characteristic edges and texture may provide better predictive features.
>
> $\textbf{A more nuanced discussion of what makes a feature spurious or not:}$
> We thank the reviewer for pointing this out. Placing attention on the background is commonly considered 'spurious' information in interpretable deep learning methods for computer vision tasks, with the original ProtoPNet paper doing so in Section 2.6 (Chen et al. 2019). Our understanding is that the reason for this trend is a desire for image classification systems to make decisions not on correlative features but causal ones. While the background of the image being a blue sky raises the probability of the bird in the image being a swallow and not a woodland thrush, the presence of the blue sky does not make the swallow a swallow. We believe this focus on wanting to learn causal and not correlative features comes from the want for interpretable methods like ProtoPNet to generalize across image classification tasks.
>
> $\textbf{The application of improving interpretability and high-level understanding about RLHF, ProtoPNets, or general neural networks: }$
> As we have discussed, the use of the CUB-200-2011 dataset is largely a demonstration of how one could use ProtoPNet and R3-ProtoPNet to perform image classification according to a particular reasoning framework. A limitation of this work is that we only propose a hypothetical application, but do not provide the reader with a particular application where we work with domain experts to solve a particular problem. The questions of who this is specifically for, and what specific purpose is this for, are left unanswered by our work. We hope to remedy this limitation in future work.
>
> That said, as discussed earlier, we believe our method has general applicability to interpretable deep learning. As we demonstrate, RLHF or related ideas have the potential to simplify the process for designing and training interpretable deep methods, with possible follow-ups extending beyond the architecture of ProtoPNet.
>
> We again thank the reviewer for these helpful comments!

---

### Author Response · Authors · 2023-11-22

We'd like to thank all the reviewers for their constructive feedback. This is just a reminder that our modified paper has been submitted, and we are looking forward to any comments regarding our rebuttals or the modified version of the paper.

---

### Meta-Review · Area_Chair_4dLC · 2023-12-06

**Metareview:**

The paper proposes R3-ProtoPNet, an extension of the prototypical part network (ProtoPNet) using Reinforcement Learning from Human Feedback (RLHF).The approach involves reweighing, reselecting, and retraining using human-rated prototype quality. Experiments were conducted on the CUB-200 dataset, showing notable improvements in prototype quality and interpretability.

Strengths:
- Clear and effective presentation
- Empirical improvements in interpretability and accuracy over standard ProtoPNet

Weaknesses:
- Novelty is fairly incremental, combining RLHF and ProtoPNet
- Findings are limited to a single dataset (CUB-200).
- Insufficient empirical investigation overall

**Justification For Why Not Higher Score:**

See weaknesses above.

**Justification For Why Not Lower Score:**

N/A

---

### Decision · Program_Chairs · 2024-01-16

Reject